# Effect of Rest Period Duration between Sets of Repeated Sprint Skating Ability Test on the Skating Ability of Ice Hockey Players

**DOI:** 10.3390/ijerph182010591

**Published:** 2021-10-09

**Authors:** Jakub Baron, Subir Gupta, Anna Bieniec, Grzegorz Klich, Tomasz Gabrys, Andrzej Szymon Swinarew, Karel Svatora, Arkadiusz Stanula

**Affiliations:** 1Institute of Sport Sciences, Jerzy Kukuczka Academy of Physical Education, 40-065 Katowice, Poland; j.baron@awf.katowice.pl (J.B.); a.bieniec@awf.katowice.pl (A.B.); 2Faculty of Medical Sciences, University of West Indies, Cave Hill 11000, Barbados; subir.gupta@cavehill.uwi.edu; 3Zagłębie Sosnowiec Hockey Club, 41-200 Sosnowiec, Poland; g.klich@onet.pl; 4Department of Physical Education and Sport, Faculty of Education, University of West Bohemia, 30100 Pilsen, Czech Republic; tomaszek1960@o2.pl (T.G.); svatorak@icloud.com (K.S.); 5Faculty of Science and Technology, University of Silesia in Katowice, 41-500 Chorzów, Poland; andrzej.swinarew@us.edu.pl; 6Department of Kinanthropology and Humanities, Faculty of Physical Education and Sport, Charles University, 16252 Prague, Czech Republic

**Keywords:** average speed, blood lactate concentration, rate of perceived exertion, speed decrement, average heart rate, peak heart rate

## Abstract

The aim of this study was to determine the effects of two different rest periods, 2 min and 3 min, between consecutive sets of a repeated sprint skating ability (RSSA) test, on the skating ability of ice hockey players. Two RSSA tests, RSSA-2 and RSSA-3, were assessed on 24 ice hockey players. In RSSA-2, six sets of 3 × 80 m sprint skating, with 2 min passive recovery between two consecutive sets was allowed. In RSSA-3, the recovery period between the sets was 3 min. Average speed, average heart rate (HR_aver_), peak heart rate (HR_peak_), blood lactate concentration ([BLa]), and rate of perceived exertion (RPE) were measured in both RSSA-2 and RSSA-3 tests. In all the sets, except set 1, the average speed of the subjects was significantly (*p* < 0.05) higher in RSSA-3 than the respective set in RSSA-2. Average HR and RPE were higher in RSSA-2 than RSSA-3 in most of the sets. For any given set, no difference in HR_peak_ was noted between RSSA-2 and RSSA-3. Post-sprint (Set 6) [BLa] was significantly (*p* < 0.05) higher in RSSA-3 than RSSA-2. This study concludes that the 3 min rest period is more beneficial than the 2 min rest period, for (1) increasing skating speed and (2) reducing overall cardiac workload and perceived fatigue.

## 1. Introduction

Ice hockey is a high-intensity intermittent team game, characterized by rapid changes in skating speed and direction, and frequent high-impact body contact. The actual time of play of ice hockey is 60 minutes, which is divided in three 20 minute periods. However, total time of match play often extends over 3 hours, including two rest intervals [1,2]. Ice hockey players are substituted frequently to keep the speed of the game very fast [3]. The effective or actual time an ice hockey player plays averages 15 to 24 min, which is much shorter than players of most other team sports [3]. Players in a typical ice hockey match alternate at nearly regular intervals, called a “shift”. The duration of each shift usually varies from 30 to 80 s (average 45 s) with 2 to 5 min of rest between shifts [3].

In the game of ice hockey, sprint skating performed is frequently interspersed with brief recovery intervals. Repeated sprint skating leads to fatigue that does not allow the player to sustain sprint. The ability to produce power quickly during brief intervals determines the successful performance of repeated sprints by the players. Quicker regeneration of energy between bouts of sprint skating during a match play gives a hockey player an edge over his/her opponent [4,5,6].

A number of repeated sprint ability (RSA) tests have been designed for players of team sports, depending on the demands of an intense period of competitive match play. Fatigue index, performance decrement index or speed decrement are often used to express the drop-off in performance in an RSA test [7]. In spite of an exponential increase in interest, the physiological mechanism responsible for drop-off performance in RSA-test is unclear [8]. However, RSA performance can help in implementing appropriate training, if required, and thus improve match performance by delaying fatigue [8].

The purpose of quick and regular substitution of ice hockey players during a match play is to play the game faster and attack or defend the opponent players with bouts of faster movement. The duration of the rest period between the shifts of play is determined by the team coach based on the match strategy and also on the recovery of anaerobic energy that was utilized during the shift of play [9]. However, to the best of our knowledge, no such study was conducted that determined the effect of the rest period duration between shifts of play or in an appropriately designed RSSA test. As 2 to 5 min rest periods are commonly used between shifts of play in hockey match play, our effort in this study was to compare the effects of two different durations, 2 min and 3 min between the bouts of RSSA tests, on the performance and psychophysiological response of the ice hockey players. This study attempted to mimic a few movement patterns and rest periods of ice hockey match play with a repeated sprint skating ability (RSSA) protocol. The major objective of this study was to determine the effects of two different durations of rest period (passive recovery), 2 min and 3 min, between two consecutive sets of RSSA tests. The effects of rest period duration on performance were determined by analysing average skating speed, speed decrement (S_dec_), heart rate (HR), blood lactate concentration ([BLa]), and rate of perceived exertion (RPE) at relevant times of the RSSA tests. It is hypothesized that the 3 min rest period between the sets of the RSSA (i.e., RSSA-3) test is advantageous over the 2 min rest period between the sets (i.e., RSSA-2), for (a) increasing the average skating speed and [BLa] but without rise in and drop-off performance (i.e., S_dec_) and imposing excess workload on the heart and (b) reducing perceived exertion.

## 2. Materials and Methods

### 2.1. Subjects

Twenty-four professional outfield ice hockey players (age: 22.65 ± 4.77 years, ranging from 17 to 34 years of age; height: 181.3 ± 4.1 cm; weight: 81.5 ± 7.2 kg), from a top division club in southern Poland, served as subjects. They had 5 to 13 years of experience of playing competitive ice hockey.

### 2.2. Main Test Procedures

#### 2.2.1. Study Design

The study was conducted in the pre-season of the year 2020–2021. During the off-season, prior to the pre-season, all the players were engaged in a periodized strength and conditioning program. The whole study was conducted over 4 non-consecutive days. On day 1, height, body weight, and body composition of the subjects were determined. Skating multistage aerobic test (SMAT), for the measurement of maximum heart rate (HR_max_) and VO_2max_, was conducted on day 2. Repeated sprint skating ability tests, using two different durations of rest period (i.e., RSSA-2, using 2 min rest period, and RSSA-3, using 3 min rest period) were determined on days 3 and 4, in a randomized order. Experiments on days 2, 3, and 4 were conducted under similar experimental condition and at the same time of the day (10 a.m. to 1 p.m. and 6 p.m. to 9 p.m.). Day 3 was separated from day 2 by 4 days in all the subjects. Subjects were instructed to maintain normal diet and fluid intake and to abstain from heavy physical activity 24 h before experiment. The experimental protocol and potential risks involved in the study were described to the subjects before giving written informed consent. The study was approved by the Ethics Committee of the Jerzy Kukuczka Academy of Physical Education in Katowice (approval number: 8/2018).

#### 2.2.2. Measurement of Body Composition

Body weight and height, total muscle mass, fat mass, and total body water were measured by body composition analyzer (InBody170, Biospace, Seoul, Korea). This is a non-invasive device that uses bioelectrical impedance analysis (BIA) method by using multi-frequencies and eight-point tactile electrodes to measure body composition.

##### Skating Multistage Aerobic Test (SMAT) for the Measurement of Maximum Heart Rate and Prediction of VO_2max_

This on-ice test for the prediction of VO_2max_ was designed by Leone et al (2007) [10]. This test was conducted on a 45 m course defined with markers at both extremities of the indoor ice hockey rink. The subject skated, holding the hockey stick with a preferred hand, from one end to the other with a predefined speed and then the speed was gradually increased until he was unable to maintain it any more. The pace was dictated by audible signals emitted by a calibrated audio player. The initial skating speed was set at 3.5 m∙s^−1^, which was increased stepwise by 0.2 m∙s^−1^. A 30 s rest period was allowed before beginning the next stage. No further rise of HR at the most exhaustive stage of the test indicated attainment of HR_max_ in the subject [11]. VO_2max_ was estimated from the table based on the formula [10]:VO_2max_ = 18.07 × (maximum velocity in m∙s^−1^) − 35.596 mL · kg^−1^ · min^−1^(1)

##### Warm-up before RSSA-2 and RSSA-3 Tests

The warm-up procedure on each day of the RSSA tests was similar. The total duration of the warm-up was 20 min, which consisted of 15 min and 5 min of off-ice and on-ice warm-ups, respectively, separated by about 20 min passive break to put on hockey gear. The off-ice warm-up was done with the use of the RAMP (raise-activate-mobilise-potentiate) protocol [12]. The on-ice warm up consisted an alternating interplay of fast and slow skating for preparing for the RSSA-2 and RSSA-3 tests.

##### Repeated Sprint Skating Ability Test

This test was originally designed by Hůlka et al. [13], based on the work rate profiles e.g., movement pattern and duration of each sprint of the players found during match play. In this test, the subjects were required to perform 6 sets of 3 × 80-m sprint trials. Subjects abstained from heavy training during the last 24 h before the RSSA test. Each set of sprint trials consisted of three repetitions of 80 m sprints, where two consecutive repetitions were spaced by 30 s of passive recovery period. Figure 1a illustrates the direction of skating movement of each repetition of RSSA-2 and RSSA-3 tests. In each of the repetitions in a set, a player has to skate forward and backward, makes a stoppage, and makes a sharp turn at some stage of the 80 m total distance. Each sprint consists of 18 m of skating forward straight from the goal line, stopping at blue line (2-m from barrier), followed by skating backward 22 m on the goal line, then skating forward 22 m and finished by sharp right on the blue line and the last 18 m skating forward to finish at the goal line. The subject has to go back at the starting point by skating (gliding) slowly. The total recovery time between two sprints, i.e., the time spent from the end of one sprint to the beginning of the next is 30 s (including the slow skating time from the end point of one sprint trial to the beginning of the next repetition).

The sprint time was measured by using photocell automatic laser timing system (Microgate, Racetime 2, Bolzano, Italy). Recovery time was measured by using a hand stopwatch. Based on the rest period (bench time) between two sets, the subjects performed two types of RSSA test (RSSA-2 and RSSA-3), separated by 5 days, in a randomized order. The overall study protocol has been presented schematically in Figure 1b.

RSSA-2: In this test, the rest period or bench time between two sets was 2 min.

RSSA-3: The rest period between two consecutive sets was 3 min.

##### Speed Decrement (S_dec_)

Speed decrement for any given set of RSSA-2 and RSSA-3 tests was calculated by using the following formula [14]:S_dec_ (%) = [(S1 + S2 + S3)/(3 × S_best_) − 1] × 100(2)
where S1 to S3 are times of 3 sprint repetitions in a set and S_best_ is the best sprint time of the given set.

##### Recording of Heart Rate

Heart rate was recorded continuously by heart rate telemeter Polar Team 2, during the whole period of the SMAT, at an interval of 2 s. In the cases of the RSSA-2 and RSSA-3 tests, HR was recorded at every 2 s, from the beginning of the warm-up till the end of the last (6th) set. The highest HR attained during a set of RSSA-2 and RSSA-3 tests was noted as the peak HR (HR_peak_) and the lowest HR recorded was noted as the minimum HR (HR_min_) of the subject. The minimum HR was noted in order to examine the extent of resetting of the cardiovascular system and recovery before the next set of the RSSA test begins. The average or mean HR (HR_aver_) was calculated from the recorded HR during the entire period of a set (i.e., from the beginning of repetition 1 to the end of repetition 3).

##### Measurement of Blood Lactate Concentration

Finger-prick capillary blood samples were collected by health professionals for the measurement of blood lactate by automated lactate analyzer (Biosen C-Line, EKF Diagnostics, UK). This determines the blood lactate by enzymatic-amperometric method using chip-sensor technology. Blood samples were collected within 1 min after the end of on-ice warm up and 3 min following the end of the last (6th) set of RSSA-2 and RSSA-3 tests. Precautions were taken to avoid possible dilution of the blood sample by tissue fluid and perspiration. Blood samples were analyzed for [BLa] within 6 h of blood withdrawal.

##### Rate of Perceived Exertion

The RPE was determined by using the Borg’s CR-10 scale [15]. The subjects indicated their RPE at rest, immediately after warm-up, and after the end of each set of RSSA-2 and RSSA-3 tests.

### 2.3. Statistical Analyses

Mean and standard deviation (SD) were used to represent the average and the typical spread of values of all measurable variables. The Normal Gaussian distribution of the data was verified by the Shapiro–Wilk test. Homoscedasticity and sphericity of data were tested by Levene’s and Mauchly’s tests, respectively. A two-way analysis of variance with repeated measures and Tukey’s HSD (honestly significant difference) post hoc was used to investigate differences in variables. In relation to the results obtained on the basis of the Borg’s CR-10 scale, Friedman’s analysis of variance and Dunn’s post hoc tests were used. The effect size (ES) of the intervention was calculated using Cohen’s guidelines. Threshold values for ES were >0.2 (small), >0.6 (moderate), >1.2 (large), and >2.0 (very large) [16]. The relationship between HR_aver_ and RPE, and between average speed in the last set of RSSA and post-RSSA blood lactate, were determined with Pearson’s product-moment correlation analysis. Statistical significance was set at *p* ≤ 0.05. All statistical analyses were conducted using Statistica 13.3 (TIBCO Software Inc., Palo Alto, CA, USA).

## 3. Results

### 3.1. Physical and Physiological Characteristics

Age, height, body mass, body fat percentage, predicted VO_2max_, and HR_max_ of the subjects are presented in Table 1. The VO_2max_ and HR_max_ were determined by SMAT.

### 3.2. Repeated Sprint Skating Ability Test Performance

Figure 2 presents the average skating speed of the subjects, in both RSSA-2 and RSSA-3 tests. All the sets of RSSA-3 were performed faster than the corresponding set of RSSA-2 and the difference was significant (*p* < 0.05) from Set 2 through 6. Skating speed was fastest in the Set 1 of RSSA-2 and then declined in the subsequent sets although the difference (*p* < 0.05) was not significant between all the consecutive sets. In RSSA-3, on the other hand, no significant decline in skating speed was observed between any two given sets.

### 3.3. Speed Decrement

Speed decrement (%) of the subjects, both in RSSA-2 and RSSA-3, has been presented in Figure 3. The S_dec_ is higher in RSSA-2 than RSSA-3 although no significant difference was noted between the two.

### 3.4. Heart Rate Response

Average-, minimum-, and peak HR (HR_peak_) of the subjects during RSSA-2 and RSSA-3 tests are presented in Figure 4. In the RSSA-2 test, HR_aver_ of the subjects varied from 156.4 ± 10.4 beats/min (80.3 ± 5.1% of HR_max_) in Set 1 to 172.3 ± 6.8 beats/min (88.7 ± 3.6% of HR_max_) in Set 6, whereas in RSSA-3, the respective values are 155.5 ± 8.1 and 166.7 ± 7 beats/min (80.1 ± 4.9% and 86.0 ± 3.8% of HR_max_). Average HR is almost similar in Set 1 of RSSA-2 and RSSA-3, but higher in rest of the sets of RSSA-2 than the corresponding set of RSSA-3, although significant difference (*p* < 0.05) was noted in Sets 3, 4, 5, and 6 only.

Like HR_aver_, a gradual increase in HR_min_ and HR_peak_ was noted in both RSSA-2 and RSSA-3 tests. Minimum HR of the subjects, attained just before the beginning of each set, was significantly higher in RSSA-2 than the corresponding HR_min_ in RSSA-3, excepting in Set 1. Peak HR, on the other hand, is similar in both RSSA-2 and RSSA-3 and reaches nearly plateau after Set 2, and no significant difference was observed between the two. The lowest and the highest HR_peak_ in RSSA-2 were 173.9 ± 10.5 beats/min or 89.3 ± 5.0% of HR_max_ (in Set 1) and 182.4 ± 6.3 beats/min or 93.8 ± 3.6% of HR_max_ (in Set 6) respectively. Similarly, the lowest- and the highest HR_peak_ attained in RSSA-3 were 173.3 ± 6.8 beats/min or 89.1 ± 3.6% (in Set 1) and 180.5 ± 5.9 beats/min or 92.8 ± 3.5% HR_max_, respectively.

### 3.5. Rate of Perceived Exertion

Figure 5 shows the RPE recorded after each set of RSSA-2 and RSSA-3 tests. A gradual increase in the RPE was found in both RSSA-2 and RSSA-3. The RPE was found significantly higher in RSSA-2 than the corresponding set of RSSA-3, except in set 6.

### 3.6. Blood Lactate

The [BLa] after warm-up and 3 min following the completion of the last set of both RSSA-2 and RSSA-3 tests are presented in Figure 6. Higher [BLa] after RSSA-3 (12.27 ± 2.75 mmol/L) was noticed in comparison to RSSA-2 (8.72 ± 2.59 mmol/L). No significant difference between warm up [BLa] before RSSA-2 (1.98 ± 0.68 mmol/L) and RSSA-3 (2.31 ± 1.01 mmol/L) was found.

## 4. Discussion

The key findings of this investigation are that RSSA-3 causes the players to skate faster, reduces HR_aver_ and RPE, and increases lactate production, in comparison to RSSA-2 test.

Ice hockey players need to develop anaerobic endurance, power, and muscle strength to match the high-intensity bursts of energy output during games. The VO_2max_ of the players are very similar to the top Polish ice hockey players [5] but lower than elite ice hockey players [17]. A lean body composition is essential for optimization of their performance as well as to support fast skating, quick change of direction, balance, agility, and frequent high impact body contact [18,19]. Lower division players were reported to contain more body fat than higher division players of NCAA [4].

Results clearly indicate that one extra minute of rest period made a significant difference in average speed, HR, and perceptual responses in the players. An important fitness requirement of an ice hockey player is the ability to recover early and to reproduce performance in subsequent skating sprints [14]. Phosphocreatine is the most important and immediate source of energy for repeated sprint exercise. During a single short sprint, the contribution of the oxidative phosphorylation is limited (<10%) [20,21]. However, with the repetitions of the sprint, the level of aerobic ATP contribution increases progressively and may share up to 40% of the total energy supply in the final repetitions of a repetitive sprint exercise [20].

A large depletion of phosphocreatine occurs immediately after repeated sprint exercise, and longer than 5 min may be required for complete recovery of the phosphocreatine back to the pre-exercise level [22,23]. Thus, only partial restoration of phosphocreatine occurs during the recovery periods between the sets of RSSA-2 and RSSA-3 tests. Gradual reduction of phosphocreatine at the beginning of subsequent sets of RSSA may be an important cause of sprint decrement in the players. Higher skating speed and lower S_dec_ in RSSA-3 possibly results from greater restoration of phosphocreatine due to longer rest period.

Blood lactate in ice hockey match play varies from 8.2 mmol/L to 13.7 mmol/L [24]. The [BLa] recorded in RSSA-2 and RSSA-3 tests of this study show similarities with the ice hockey match play. In RSSA-2, the [BLa] ranged from 5.58 to 14.99 mmol/L, with the mean value of 8.72 mmol/L. In RSSA-3, the [BLa] reached 12.27 (±2.75) mmol/L at the end. The mean [BLa] in RSSA-3 was 12.27 ± 2.75 mmol/L that ranged from 6.16 to 17.71 mmol/L. Higher skating speed in RSSA-3 is probably responsible for higher [BLa] in spite of longer rest period in comparison to RSSA-2. Studies [25] have shown that the contribution of the anaerobic glycolysis in the initial sprint is high but inhibited progressively as sprints are repeated. The relationship between the anaerobic glycolytic rate and repeated sprint performance, however, is still unclear [14,25].

Ice hockey matches are played with HR_aver_ of about 85% of HR_max_ and the HR often exceeds 90% of HR_max_ [18,26,27]. A steady rise in HR with the progression of sets of repeated sprint tests confirms the involvement of a greater aerobic contribution during the latter sets. The maximal workload on the heart, indicated by HR_peak_, in both RSSA-2 and RSSA-3, is similar as indicated by similar HR_peak_ although the players experienced higher overall cardiac workload in RSSA-2 than in RSSA-3. This is because of higher HR_min_ in RSSA-2. Shorter rest period in RSSA-2 causes the HR remains elevated. This may be caused by higher circulating catecholamines in the beginning of a set of RSSA-2 than the corresponding set of RSSA-3 [28]. A rise in HR before the start of any short-burst of high intensity exercise prepares the individual for faster circulatory adjustment and quicker release of energy. However, slower HR recovery, before the next set of bouts, is likely to reduce the cardiac reserve that leads to early fatigue [29,30].

The RPE is a valid marker of exercise intensity. This is also a recognized means of measuring training effects using standardized exercise test protocols [31]. The RPE enables an individual to evaluate how hard or easy a physical task is at any point of time by integrating afferent stimuli and feedforward mechanisms [32]. The RSSA-3 in this study induced significantly lower RPE in spite of higher average skating velocity when compared with RSSA-2. Lower HR_aver_ and longer rest period likely to be responsible for reduced perception of fatigue in RSSA-3, although HR_peak_ in both the protocols were similar.

The usual rest period between the shifts during ice hockey match play varies widely. In this study we tried to compare some differences in physiological and perceptual response between 2- and 3 min rest periods while performing an RSSA test that was based on some basic movement patters of ice hockey players during real match play. Substitution of the players is likely to depend largely on the match strategy by the coach and partly on the physiological ability of the players. More studies are required that can evaluate the difference in physiological response between rest periods of other durations, e.g., between 2- and 4 min, 3- and 5 min, or 2- and 5 min. However, a 3 min rest period is advantageous over the 2 min rest duration where the coach has to decide the option of a rest period from 2- or 3 min. Thus, this study cannot conclude when the choice of rest period is not 2 min and 3 min but longer durations like 4 min and 5 min.

Only one withdrawal of blood for the measurement of [BLa] following sprint skating is a limitation of this study. Measurement of [BLa] after every set of RSSA tests could be more useful to determine the role of anaerobic glycolysis and removal pattern of lactate from blood. However, lactate measurement after every set of RSSA tests is meaningful when the rest period is preferably 4 min or longer because [BLa] reaches its peak value only after 3 min following the end of sprint events [33].

## 5. Conclusions

The results of the present study suggest the potential benefits of a 3 min rest period (RSSA-3) over a 2 min rest period (RSSA-2) for (1) increasing the average skating speed, (2) reducing overall cardiac workload, and (3) reducing the perception of fatigue. This study may have practical importance to the ice hockey coaches for effective use of their players by considering duration of bench time between shifts of play.

## Figures and Tables

**Figure 1 ijerph-18-10591-f001:**
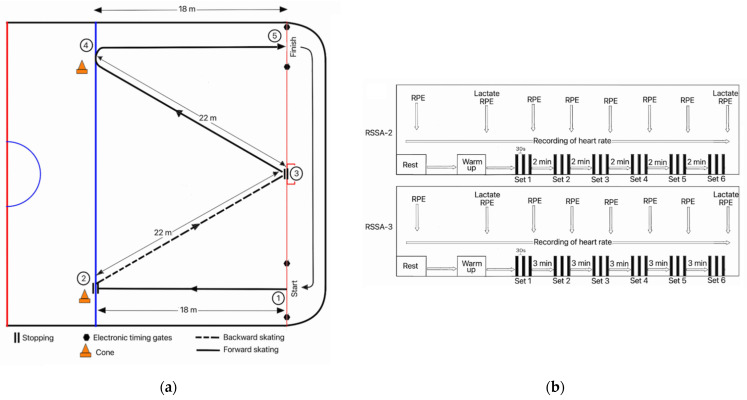
Study protocol. (**a**) Diagrammatic presentation of the direction and distance of skating in the repeated sprint skating ability test. (**b**) Schematic representation of the study protocol. This shows six sets of repeated sprint skating ability tests with two different passive rest period durations. (1) RSSA-2: Rest period of 2 min and (2) RSSA-3: Rest period of 3 min. Down arrows indicate time for collection of blood samples and recording of RPE. Heart rate was recorded continuously from rest until the end of the RSSA test.

**Figure 2 ijerph-18-10591-f002:**
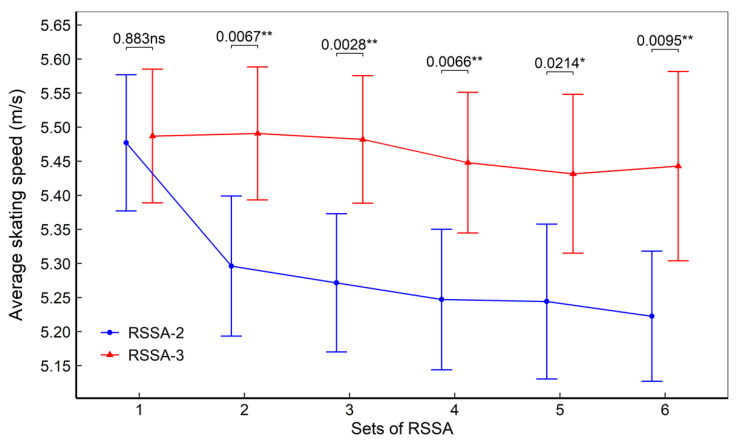
Average skating speed during RSSA-2 and RSSA-3 tests (ns—non-significant, ** *p* < 0.01; * *p* < 0.05).

**Figure 3 ijerph-18-10591-f003:**
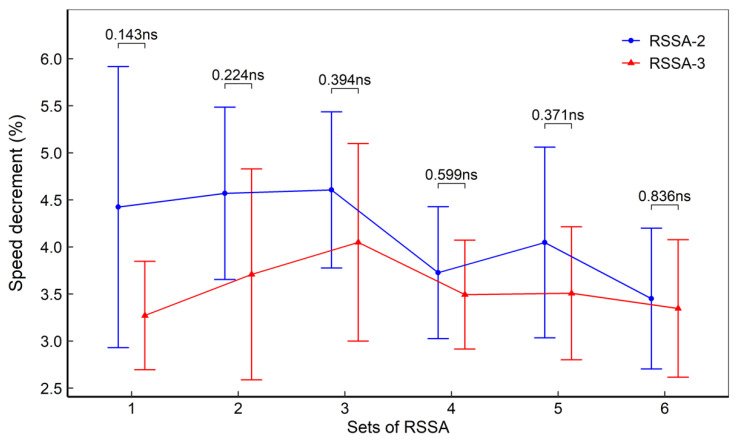
Speed decrement during RSSA-2 and RSSA-3 tests (ns—non-significant).

**Figure 4 ijerph-18-10591-f004:**
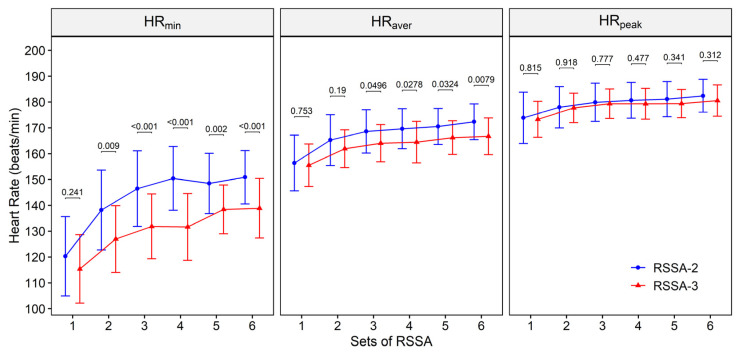
Heart rate response during RSSA-2 and RSSA-3 tests. P values between corresponding sets of RSSA-2 and RSSA-3 have been indicated above upper bars of 95% CI.

**Figure 5 ijerph-18-10591-f005:**
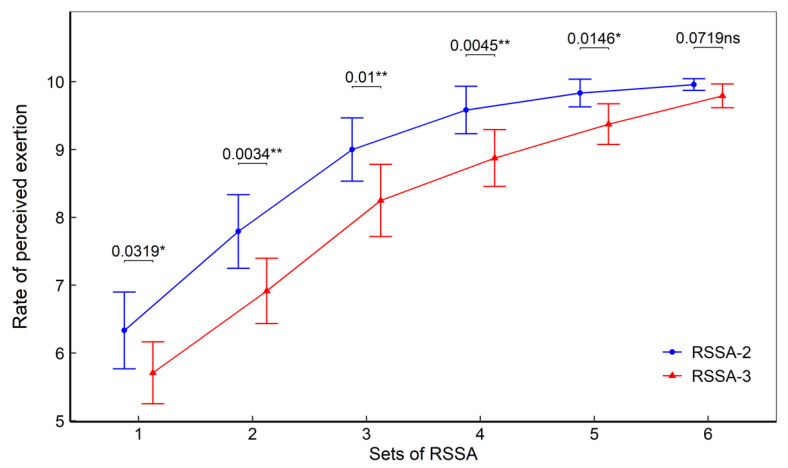
Rate of perceived exertion in RSSA-2 and RSSA-3 tests (** *p* < 0.01; **p* < 0.05, ns—non-significant).

**Figure 6 ijerph-18-10591-f006:**
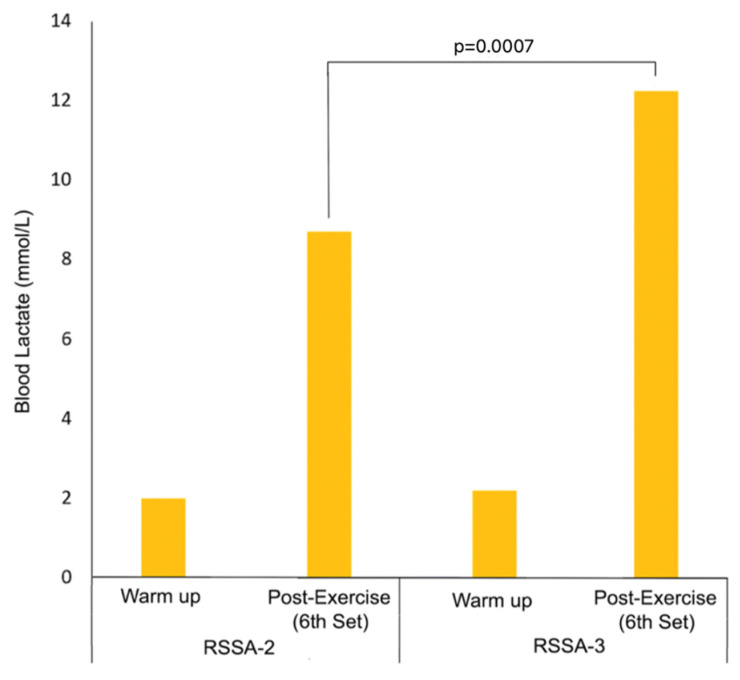
Blood lactate concentration after warmup and following the last set of RSSA-2 and RSSA-3 tests.

**Table 1 ijerph-18-10591-t001:** Physical characteristics of the volunteers (*n* = 24).

Age (years)	Height (cm)	Weight (kg)	Body Fat (%)	Muscle Mass (kg)	Predicted VO_2max_ (mL·kg^−1^·min)	HR_max_ (beats·min^−1^)
22.65 ± 4.77	181.3 ± 4.2	81.5 ± 7.4	15.4 ± 3.9	39.5 ± 3.3	51.7 ± 4.2	197.2 ± 11.1

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
