# Peer review of "Effect of Rest Period Duration between Sets of Repeated Sprint Skating Ability Test on the Skating Ability of Ice Hockey Players"

_ijerph, 2021, doi:10.3390/ijerph182010591_

Round 1
Reviewer 1 Report
Dear authors, congratulations on your paper, I see you have taken into consideration some of my comment but not all of them, here you have my previous comments in regular letters and my new comments in bold letters
Please look at the way the journal explain how to describe the figures. Please address it.
Lines 33 and 327. Lines 33 and 365 in the new document. I would say that 3 min rest period is beneficial for in stead of by. Because 2 min rest period is beneficial to increase training density and intensity for example, but if you want to work on the speed the you should use 3 min rest periods. Anyway, it will be interesting if in the introduction you explain the actual rest periods in game for reader that are not familiar with the game, because in my opinion it is not the same bench rest that the time between bouts of exercise. Specificity is one of the most importan principle of training. This will justify your whole manuscript. I believe the author change it properly.
Line 43. “Ice hockey players are substituted more frequently than any other team game” Have you compared them with Handball players? How do you know this data? Please cite This has not been addressed, please do so.
Line 80 also should say “for” in stead of by increasing, please change it through the whole manuscript.
Line 95. I would say over 4 non-consecutive days because it seams that RSSA-2 and RSSA-3, were separated by 5 days as you say in line 145. Furthermore, I will be interesting that you mention the rest days between SMAT and RSSA. Good job regarding this comment.
Line 124 . The SMAT was design to predict V ̇ O2max. How do you measure HRmax? How do you actually know the reached their HRmax? Do you mean HRpeak? Please explain. Authors did not explain this properly. Technically, they are measuring HRpeak and they say so in the abstract, please use the same terminology during the whole text.
Line 185. What is the goal of measuring the lowest HR? You should explain that here. It will be easier to understand if you say HR after recovery. OK.
Line 266. Figure 4. The description of this figure is incorrect, please change that. Furthermore I am confuse about the x-axis since for HRmin you write through the manuscript about beats/min but I believe in the other you are talking about % of HRmax, please clarify that. Good job regarding this comment.
Line 269 I believe this is a mistake in translation, I would call the section “ Rate of perceived exertion” as you mention in Materials and Methods. Perfect.
Line 293. Do you mean Polish ice hockey players? Please clarify that. Excellent.
Line 319. Could you explain the same for both protocols so the reader can compare? ok, thank you.
Lines 333 to 336. “Rise in HR before the start of any short-burst of high intensity exercise prepares the individual for faster circulatory adjustment and quicker release of energy, but too high HR can be detrimental and causes early fatigue”. I do agree with the first sentence, but what is too high? Please cite. Good job regarding this comment.
Line 364. You forgot to erase “This section is mandatory”. The authors have forgotten this again.
Author Response
Thank you very much for your constructive feedback and remarks aiming at improving the manuscript. We hope the revised version and the responses given to the comments will meet your expectations. Please note that all changes are highlighted in yellow in the revised version of manuscript.
Reviewer’s comments: Lines 33 and 327. Lines 33 and 365 in the new document. I would say that 3 min rest period is beneficial for in stead of by. Because 2 min rest period is beneficial to increase training density and intensity for example, but if you want to work on the speed the you should use 3 min rest periods. Anyway, it will be interesting if in the introduction you explain the actual rest periods in game for reader that are not familiar with the game, because in my opinion it is not the same bench rest that the time between bouts of exercise. Specificity is one of the most importan principle of training. This will justify your whole manuscript. I believe the author change it properly.
Reply: Thanks for the comment.
We have changed (Lines 33 and 365) as suggested.
We have mentioned the rest intervals (Line 42) and the effective or actual time an ice hockey player plays (Lines 44-45), duration of each shift, and average time between shifts of play (Line 47). There is no specific time for a shift and nor there is any specific time between two shifts. All these depend on a coach’s strategy based on the match analysis and ability of his players.
Reviewer’s comment: Line 43. “Ice hockey players are substituted more frequently than any other team game” Have you compared them with Handball players? How do you know this data? Please cite This has not been addressed, please do so.
Reply: We have changed the sentence as follows:
“Ice hockey players are substituted frequently to keep the speed of the game very fast (Brocherrie et al, 2018).
Reviewer’s comment: Line 80 also should say “for” in stead of by increasing, please change it through the whole manuscript.
Reply: Thank you. We have changed the sentence according to your comment. Now the sentence reads as follows:
“……………..between the sets (i.e., RSSA-2), for (a) increasing the average…………”
Reviewer’s comment: Line 95. I would say over 4 non-consecutive days because it seams that RSSA-2 and RSSA-3, were separated by 5 days as you say in line 145. Furthermore, I will be interesting that you mention the rest days between SMAT and RSSA. Good job regarding this comment.
Reply: Thank you.
Reviewer’s comment: Line 124 . The SMAT was design to predict VO2max. How do you measure HRmax? How do you actually know the reached their HRmax? Do you mean HRpeak? Please explain. Authors did not explain this properly. Technically, they are measuring HRpeak and they say so in the abstract, please use the same terminology during the whole text.
Reply: We measured not only VO2max by the SMAT but also HRmax. Heart rate, in most of the cases in this manuscript, were expressed in absolute values (i.e., beats/min). However, to give a clear picture to the readers of this article, we have expressed HRmin, HRaver, and HRpeak values in terms of HRmax% at many places in this manuscript (for example, lines 249, 251, 259-262, 326-326). Expressing HR in terms of HRmax is a more rational way to compare HR response of the participants among various age groups and different categories of participants.
Besides VO2max, HRmax has been measured successfully by multistage shuttle run test (Paradisis et al, 2014) and SMAT (Allisse M, Bui HT, Léger L, Comtois AS, Leone M. Updating the Skating Multistage Aerobic Test and Correction for V[Combining Dot Above]O2max Prediction Using a New Skating Economy Index in Elite Youth Ice Hockey Players. J Strength Cond Res. 2020 Nov;34(11):3182-3189. doi: 10.1519/JSC.0000000000002602. PMID: 33105369). We have added Allisse et al 2018 as a reference in the text (Line 127).
Reviewer’s comment: Line 185. What is the goal of measuring the lowest HR? You should explain that here. It will be easier to understand if you say HR after recovery. OK.
Reply: Thank you
Line 266. Figure 4. The description of this figure is incorrect, please change that. Furthermore I am confuse about the x-axis since for HRmin you write through the manuscript about beats/min but I believe in the other you are talking about % of HRmax, please clarify that. Good job regarding this comment.
Reply: Thank you
Line 269 I believe this is a mistake in translation, I would call the section “ Rate of perceived exertion” as you mention in Materials and Methods. Perfect.
Reply: Thank you
Line 293. Do you mean Polish ice hockey players? Please clarify that. Excellent.
Reply: Thank you
Line 319. Could you explain the same for both protocols so the reader can compare? ok, thank you.
Reply: Thank you.
Lines 333 to 336. “Rise in HR before the start of any short-burst of high intensity exercise prepares the individual for faster circulatory adjustment and quicker release of energy, but too high HR can be detrimental and causes early fatigue”. I do agree with the first sentence, but what is too high? Please cite. Good job regarding this comment.
Reply: Thanks for this important comment. We are adding a reference and splitting the sentence into two, as follows:
“Rise in HR before the start of any short-burst of high intensity exercise prepares the individual for faster circulatory adjustment and quicker release of energy. However, slower HR recovery, before the next set of bouts of sprint, likely to reduce the cardiac reserve function that leads to early fatigue. (Ref: Le Meur et al, 2017. Assessing overreaching with heart rate recovery: What is minimal intensity required).”
Reviewer’s comment: Line 364. You forgot to erase “This section is mandatory”. The authors have forgotten this again.
Reply: Thank you for pointing out the mistake. We have removed “This section is mandatory.”

Reviewer 2 Report
The authors improved the article according to the comments previously made.
Author Response
We thank the Reviewer for their invested time in reviewing our paper, and providing useful critical and constructive feedback.
This manuscript is a resubmission of an earlier submission. The following is a list of the peer review reports and author responses from that submission.
Round 1
Reviewer 1 Report
Dear authors, congratulations on your paper, here you have some comments that I hope can improve you work.
Please look at the way the journal explain how to describe the figures.
Lines 33 and 327. I would say that 3 min rest period is beneficial for in stead of by. Because 2 min rest period is beneficial to increase training density and intensity for example, but if you want to work on the speed the you should use 3 min rest periods. Anyway, it will be interesting if in the introduction you explain the actual rest periods in game for reader that are not familiar with the game, because in my opinion it is not the same bench rest that the time between bouts of exercise. Specificity is one of the most importan principle of training. This will justify your whole manuscript.
Line 43. “Ice hockey players are substituted more frequently than any other team game” Have you compared them with Handball players? How do you know this data? Please cite
Line 85 I would say over 4 non-consecutive days because it seams that RSSA-2 and RSSA-3, were separated by 5 days as you say in line 145. Furthermore, I will be interesting that you mention the rest days between SMAT and RSSA.
Line 105. The SMAT was design to predict V ̇ O2max. How do you measure HRmax? How do you actually know the reached their HRmax? Do you mean HRpeak? Please explain.
Line 171. What is the goal of measuring the lowest HR? You should explain that here. It will be easier to understand if you say HR after recovery.
Line 247. Figure 4. The description of this figure is incorrect, please change that. Furthermore I am confuse about the x-axis since for HRmin you write through the manuscript about beats/min but I believe in the other you are talking about % of HRmax, please clarify that.
Line 248 I believe this is a mistake in translation, I would call the section “ Rate of perceived exertion” as you mention in Materials and Methods.
Line 261 I am not sure what you mean by “post-sprint” You have not explain it in Materials and Methods.
Line 274. Do you mean Polish ice hockey players? Please clarify that.
Lines 298 and 299. Could you explain the same for both protocols so the reader can compare?
Lines 313 to 315 “Rise in HR before the start of any short-burst of high intensity exercise prepares the individual for faster circulatory adjustment and quicker release of energy, but too high HR can be detrimental and causes early fatigue”. I do agree with the first sentence, but what is too high? Please cite.
Line 326. You forgot to erase “This section is mandatory”
Kind regards,
Reviewer 2 Report
Dear reviewers,
Thank you for the opportunity to evaluate your paper, but I feel that it should be rejected for the following reasons:
There is a lack of further theoretical and conceptual development in the introduction and discussion.
The p-values should be presented exactly and not p< 0.05.
The figures are in very poor quality which makes it impossible to evaluate them.
Many sentences in the introduction and discussion do not have bibliographic references to support these statements.
The English language should be revised.
In the discussion, the limitations, practical applicability and future studies to be developed are not presented.